# Malnutrition enteropathy in Zambian and Zimbabwean children with severe acute malnutrition: A multi-arm randomized phase II trial

Malnutrition underlies almost half of all child deaths globally. Severe Acute Malnutrition (SAM) carries unacceptable mortality, particularly if accompanied by infection or medical complications, including enteropathy. We evaluated four interventions for malnutrition enteropathy in a multi-centre phase II multi-arm trial in Zambia and Zimbabwe and completed in 2021. The purpose of this trial was to identify therapies which could be taken forward into phase III trials. Children of either sex were eligible for inclusion if aged 6–59 months and hospitalised with SAM (using WHO definitions: WLZ <−3, and/or MUAC <11.5 cm, and/or bilateral pedal oedema), with written, informed consent from the primary caregiver. We randomised 125 children hospitalised with complicated SAM to 14 days treatment with (i) bovine colostrum ($n = 25$), (ii) N-acetyl glucosamine ($n = 24$), (iii) subcutaneous teduglutide ($n = 26$), (iv) budesonide ($n = 25$) or (v) standard care only ($n = 25$). The primary endpoint was a composite of faecal biomarkers (myeloperoxidase, neopterin, $\alpha_1$-antitrypsin). Laboratory assessments, but not treatments, were blinded. Per-protocol analysis used ANCOVA, adjusted for baseline biomarker value, sex, oedema, HIV status, diarrhoea, weight-for-length Z-score, and study site, with pre-specified significance of $P < 0.10$. Of 143 children screened, 125 were randomised. Teduglutide reduced the primary endpoint of biomarkers of mucosal damage (effect size −0.89 (90% CI: −1.69,−0.10) $P = 0.07$), while colostrum (−0.58 (−1.4, 0.23) $P = 0.24$), N-acetyl glucosamine (−0.20 (−1.01, 0.60) $P = 0.67$), and budesonide (−0.50 (−1.33, 0.33) $P = 0.32$) had no significant effect. All interventions proved safe. This work suggests that treatment of enteropathy may be beneficial in children with complicated malnutrition. The trial was registered at ClinicalTrials.gov with the identifier NCT03716115.

Despite 17 million annual worldwide cases of childhood severe acute malnutrition (SAM), and its high associated mortality when children are hospitalized with complications[1,2], there have been few new treatments for over three decades for children with complicated SAM.

Current management follows steps in the WHO guidelines launched in 1999[3–6] but there is an acknowledged lack of evidence for many interventions[1,7] and consensus that more trials are needed. In sub-Saharan Africa, HIV has had a major impact on SAM, causing higher

✉e-mail: m.p.kelly@qmul.ac.uk

mortality during[8,9] and after admission[2,10–13], complications like persistent diarrhoea[9], and prolonged hospital admission.

Even after recovery from acute SAM, there are very common chronic consequences, including both stunting of linear growth and short- and long-term inhibition of neurodevelopmental potential[14]. The effects of chronic malnutrition upon cognitive functioning are particularly notable in the domains of attention, memory and learning, contributing to poor school performance[15] and corresponding with abnormalities on neuroimaging[16]. Such long-term consequences may not be averted by providing improved nutrition alone, as small intestinal absorption of nutrients and essential micronutrients is compromised by ongoing malnutrition enteropathy[17]. Effective treatment of malnutrition enteropathy is thus likely to have major benefits for the development of large numbers of children within resource-poor countries.

The small intestinal mucosal damage now recognised as malnutrition enteropathy was first recognised in SAM in the 1960s[18,19]. More recent studies confirm the very high frequency of malnutrition enteropathy in resource poor countries and an association between such gut inflammation and mortality in complicated SAM[20]. There is therefore considerable interest in malnutrition enteropathy, which varies from mild villus blunting and inflammation to a severe state with total villus atrophy similar to coeliac disease. Malnutrition enteropathy is characterised by severe epithelial lesions[21,22] accompanied by mucosal inflammation in the epithelium and lamina propria together with depletion of secretory cells[23]. The epithelial lesions permit microbial translocation from the gut lumen driving systemic inflammation[24,25]. Transcriptomic analysis of mucosal biopsies confirms links between inflammation, villus blunting, microbial translocation and epithelial leakiness[26].

Recognising that fresh approaches are needed to ameliorate the mucosal damage that characterises malnutrition enteropathy we evaluated four potential therapies in a multi-arm, phase II, randomised controlled trial in two tertiary hospitals in southern Africa (Lusaka, Zambia and Harare, Zimbabwe)[27]. The Therapeutic Approaches to

Malnutrition Enteropathy (TAME) trial tested the hypothesis that one or more of these therapies could reduce the severity of malnutrition enteropathy in children with SAM. Each intervention was chosen because of its potential for enhancing intestinal repair. Bovine colostrum contains abundant growth factors, including insulin-like and epidermal growth factors[28], and demonstrates efficacy in ulcerative colitis[29,30]. N-acetyl glucosamine may restore the intestinal barrier since glycosylation is deficient in SAM[31,32], and it has been shown to promote mucosal healing in inflammatory bowel disease (IBD)[33]. Teduglutide improves nutrient absorption through mucosal regeneration in intestinal failure[34,35]. Budesonide suppresses inflammation with minimal systemic exposure in IBD[36]. A broad range of endpoints was chosen to assess several domains of pathophysiology[37,38] relevant to restoring mucosal integrity. The primary endpoint, a composite of faecal biomarkers (myeloperoxidase, α1-antitrypsin, and neopterin) was chosen to reflect mucosal inflammation and loss of barrier function. Secondary endpoints were chosen to reflect enterocyte damage (plasma intestinal fatty acid-binding protein) and the systemic response to microbial translocation (lipopolysaccharide-binding protein (LBP), C-reactive protein (CRP), soluble CD163, soluble CD14), alongside anthropometric measures of nutritional recovery, death, adverse events, diarrhoea, fever and recovery from oedema.

## Results

The TAME trial was conducted in the Children's Hospital of University Teaching Hospital, Lusaka, and Sally Mugabe Hospital, Harare. The third planned site (Parirenyatwa Hospital, Harare) was closed to recruitment due to its use as a COVID-19 centre. Children hospitalised with complicated SAM were enrolled once they were considered stable and ready to transition to higher calorie intakes, to avoid anticipated high early mortality in this high-risk population. Of 143 children screened, 133 were eligible: 8 declined, leaving 125 children enrolled: 62 in Lusaka and 63 in Harare (Fig. 1, Table 1), of whom 43% were female (Table 1). Causes of ineligibility were weight <5 kg (n = 1), clinical instability (n = 1), haemoglobin <6 g/dL (n = 2), death prior to

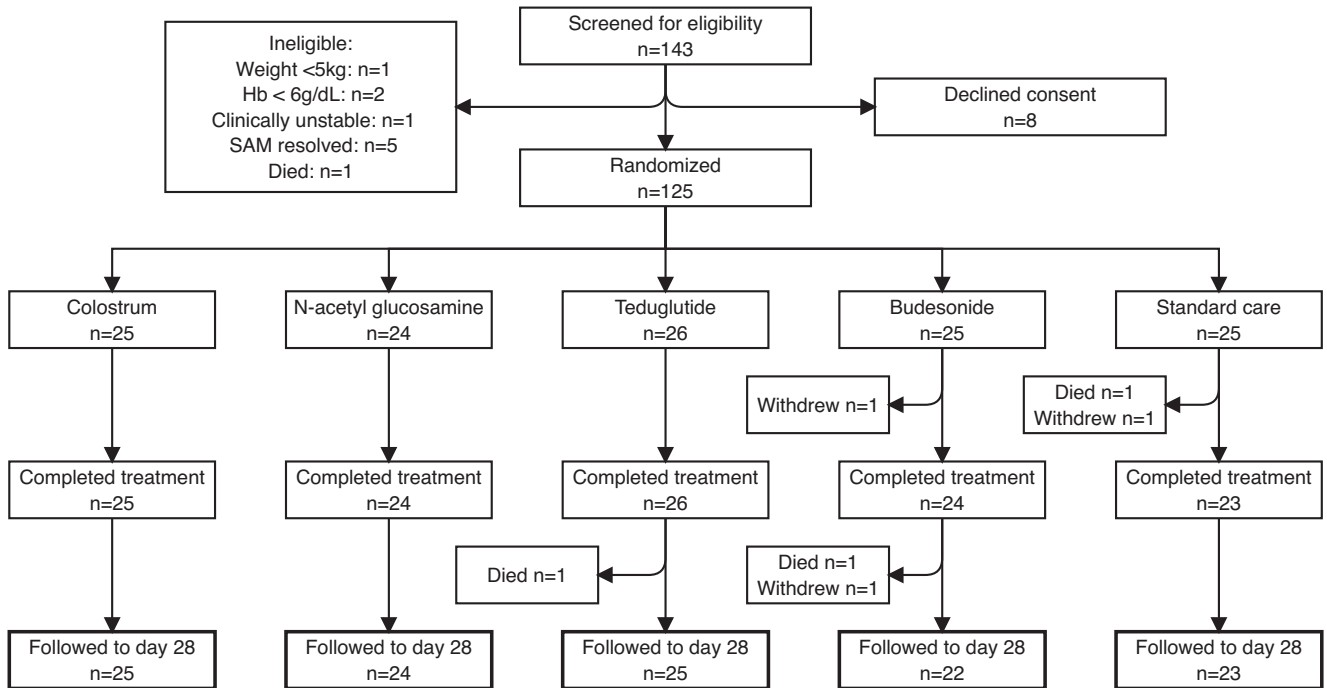

**Fig. 1 | Flowchart of participating children in the TAME trial.** One child died and two children withdrew before day 15, so day 15 endpoint data were available for 122 children for most endpoints, and 118 for the primary endpoint; all these children completed their allocated intervention and standard care and are included in the per protocol analysis. A further 2 children died and one withdrew between days 15 and 28. SAM, severe acute malnutrition. Hb, haemoglobin concentration.

**Table 1 | Clinical, demographic and household characteristics at randomisation (ie baseline, before the intervention)**

| | Colostrum N = 25 | N-acetyl glucosamine N = 24 | Teduglutide N = 26 | Budesonide N = 25 | Standard care N = 25 |
|---|---|---|---|---|---|
| Age (months) | 20 (15,23) | 19 (14,23) | 18 (12,20) | 17 (13,22) | 16 (13,27) |
| Sex, female; N (%) | 12 (48%) | 8 (33%) | 10 (38%) | 14 (56%) | 10 (40%) |
| Study site (Zambia:Zimbabwe) | 12:13 | 12:12 | 13:13 | 12:13 | 13:12 |
| **HIV status** | | | | | |
| Unexposed; N (%) | 18 (72%) | 16 (67%) | 17 (65%) | 17 (68%) | 14 (56%) |
| Exposed but uninfected | 3 (12%) | 4 (17%) | 3 (12%) | 6 (24%) | 3 (12%) |
| Positive | 4 (16%) | 4 (17%) | 6 (24%) | 2 (8%) | 8 (32%) |
| **Anthropometry** | | | | | |
| Weight-for-height (or length) Z score | −2.72 (−4.19,−2.35) | −2.59 (−3.74,−1.24) | −2.22 (−3.34,−1.44) | −2.21 (−2.69,−1.59) | −2.62 (−3.60,−1.89) |
| Height (or length)-for-Age Z score | −3.07 (−3.82,−2.72) | −4.11 (−4.63,−2.72) | −3.28 (−4.38,−2.66) | −3.04 (−3.44,−2.21) | −3.17 (−4.41,−1.98) |
| MUAC (cm) | 11.4 (10.8,11.9) | 11.5 (11.3,12.9) | 11.6 (11.0,12.3) | 12.0 (11.4,12.6) | 11.7 (11.0,12.3) |
| **Malnutrition subtype at randomisation** | | | | | |
| Oedematous | 17 (68%) | 17 (71%) | 21 (81%) | 18 (72%) | 20 (80%) |
| Non-Oedematous | 8 (32%) | 7 (29%) | 5 (19%) | 7 (28%) | 5 (20%) |
| **Clinical conditions at randomisation** | | | | | |
| Acute diarrhoea | 1 (4%) | 0 | 1 (4%) | 0 | 1 (4%) |
| Persistent diarrhoea | 0 | 0 | 1 (4%) | 1 (4%) | 1 (4%) |
| Pneumonia | 0 | 1 (4%) | 1 (4%) | 3 (12%) | 3 (12%) |
| Tuberculosis | 5 (20%) | 3 (13%) | 6 (23%) | 4 (16%) | 6 (24%) |

All continuous variables shown as median (interquartile range, IQR).

enrolment ($n = 1$), or resolution of SAM ($n = 5$). Children were randomised a median of 5.5 (range 1–21) days after admission (Table 2), following blood and stool collection to measure baseline biomarkers; no biomarker data were available from earlier in the hospital admission. One child died and two withdrew before day 15, so 122 children (98%) contributed outcome data (Fig. 1). Two further children died and one withdrew between the end of treatment and the day 28 follow-up visit. Adherence and completion were very high: 122/124 (98%) children who survived to day 15 received all planned doses.

**Primary endpoint**

The median day 15 composite faecal inflammatory score was lower in all treatment groups compared with the standard care group (Table 3). Using ANCOVA, with pre-specified P-value threshold of 0.10 and following adjustment for seven key covariates, pairwise comparisons showed the composite score in the teduglutide group was significantly lower than standard care (mean difference −0.89, 90%CI −1.69, −0.10, $P = 0.07$) (Table 4). Results were also stratified by site (Supplementary Table S1). In Harare both teduglutide and budesonide reduced the primary outcome compared to standard care, by 2.1 (90%CI 0.7, 3.5; $P = 0.02$) and 1.4 (0.06, 2.7; $P = 0.09$) respectively, but the effect was not significant in Lusaka (Supplementary Table S1). In the whole group, the ANCOVA model demonstrated no significant effect of sex, HIV infection, oedema, diarrhoea, or baseline weight-for-length Z-score (WLZ) on the faecal inflammatory score.

**Secondary endpoints**

Amongst secondary biomarker endpoints (Table 4), compared with standard care, budesonide reduced plasma CRP (mean reduction 5.2 mg/L; 90%CI 0.8, 9.6; $P = 0.05$) and sCD163 (mean reduction 405 ng/mL; 90%CI 73, 738; $P = 0.05$) while colostrum and N-acetyl glucosamine had effects only on CRP (reductions 5.9 mg/L; 90%CI 1.4, 10.3; $P = 0.03$, and 4.8 mg/L; 90%CI 0.5, 9.2; $P = 0.07$, respectively). There were notable site-specific differences in CRP, sCD14, sCD163 and iFABP (Supplementary Table S1). Transformation of secondary endpoints to approximate a normal distribution did not change the effects observed, except for colostrum, for which an effect was found on

GLP-2 only when the values were transformed (Supplementary Table S2). N-acetyl glucosamine reduced mean days with diarrhoea by 89% (Table 4; ratio of NAG to standard care 0.11; 90%CI 0.04, 0.33; $P = 0.001$). None of the interventions affected days with fever or oedema, or change in weight, height, or MUAC.

**Assessment of endoscopic biopsies**

Of 62 children enrolled in Lusaka, endoscopy was carried out on 25 (5 colostrum, 4 N-acetyl glucosamine, 6 teduglutide, 5 budesonide, 5 Standard Care). Of the remainder, 9 children were unsuitable for anaesthesia (mainly upper respiratory infections), one child had no INR result available, 3 children missed endoscopy because no anaesthetist was available; and 24 children missed endoscopy because of instrument breakdowns. Representative specimens are shown in Fig. 2. Morphometry in post-treatment biopsies showed significant differences in crypt depth (Fig. 3; $P = 0.02$ by Kruskal-Wallis test; by Dunn's test $P = 0.01$ for colostrum and $P = 0.049$ for teduglutide compared to standard care).

**Adverse events**

A total of 102 clinical adverse events were reported (10 SAEs, 92 non-serious AEs) which did not differ significantly by treatment arm (Supplementary Table S3). No AEs or SAEs were adjudicated as related to the investigative products. There were no adverse events related to endoscopy, and no Adverse Events of Special Interest. Laboratory AEs did not differ by treatment allocation (Supplementary Table S4). Of the 10 SAEs observed, 3 (2% of the whole cohort) were deaths and 7 (6%) were prolonged hospitalisations or re-admission (Supplementary Table S3). Of the three deaths, only one (on day 3) occurred before day 15; the other two occurred on days 20 and 23. Deaths were attributed to fever with diarrhoea ($n = 1$; Standard Care arm), tuberculosis ($n = 1$; Teduglutide arm), and aspiration pneumonia ($n = 1$; Colostrum arm). Seven readmissions or prolonged hospitalisations occurred, due to fever ($n = 2$), deterioration of oedema ($n = 1$), vomiting ($n = 1$), respiratory distress ($n = 1$), and a burn from a hot water bottle whilst in hospital ($n = 1$). One SAE was a readmission for observation at the day 28 visit after a protocol violation, due to receiving double the protocol

**Table 2 | Preceding conditions at or prior to admission to hospital**

| | Colostrum N = 25 | N-acetyl glucosamine N = 24 | Teduglutide N = 26 | Budesonide N = 25 | Standard care N = 25 |
|---|---|---|---|---|---|
| **Clinical conditions on admission to hospital** | | | | | |
| Acute diarrhoea | 14 (56%) | 14 (61%) | 16 (67%) | 13 (54%) | 13 (57%) |
| Persistent diarrhoea | 5 (20%) | 5 (22%) | 7 (29%) | 6 (25%) | 5 (22%) |
| Pneumonia | 5 (20%) | 4 (17%) | 4 (17%) | 6 (25%) | 3 (13%) |
| Tuberculosis | 4 (16%) | 4 (17%) | 2 (8%) | 2 (8%) | 3 (13%) |
| Urinary tract infection | 0 | 0 | 1 (4%) | 0 | 0 |
| Cerebral palsy | 0 | 2 (8%) | 2 (8%) | 2 (8%) | 0 |
| Oedema | 20 (80%) | 20 (83%) | 24 (92%) | 18 (72%) | 22 (88%) |
| **Early life nutrition** | | | | | |
| Birth weight (kg) | 3.0 (2.7,3.2) | 2.8 (2.6,3.1) | 2.9 (2.7,3.4) | 3.2 (2.8,3.4) | 3.1 (2.6,3.4) |
| Current breastfeeding; N (%) | 1 (4%) | 2 (8%) | 4 (15%) | 3 (12%) | 1 (4%) |
| Age at cessation of breastfeeding (months) | 14 (9,18) | 12 (8,15) | 14 (12,16) | 14 (10,17) | 14 (9,18) |
| **Mode of delivery** | | | | | |
| Normal Vaginal Delivery | 23 (92%) | 22 (92%) | 22 (84%) | 23 (92%) | 22 (88%) |
| Instrumental or Caesarean delivery | 2 (8%) | 2 (8%) | 3 (12%) | 2 (8%) | 3 (12%) |
| Unknown | 0 | 0 | 1 (4%) | 0 | 0 |
| **Gestational age** | | | | | |
| Preterm (<37 weeks) | 6 (24%) | 4 (17%) | 6 (24%)[a] | 4 (16%) | 2 (8%) |
| Term | 19 (76%) | 20 (83%) | 19 (76%) | 21 (84%) | 23 (92%) |
| **Duration of hospital stay prior to randomisation** (days) | 6 (5,8) | 5 (3,6) | 5 (4,7) | 4 (3,5) | 5 (3,7) |
| **Primary caregiver characteristics** | | | | | |
| Age (years) | 26 (22,32) | 28 (23,31) | 26 (21,30) | 24 (20,28) | 26 (22,30) |
| Married | 16 (64%) | 13 (54%) | 15 (58%) | 12 (48%) | 14 (56%) |
| Education (completed secondary school) | 8 (32%) | 15 (63%) | 16 (62%) | 15 (60%) | 15 (60%) |
| **Home Environment** | | | | | |
| Rural | 4 (16%) | 2 (8%) | 3 (12%) | 1 (4%) | 1 (4%) |
| Urban (high density) | 20 (80%) | 21 (88%) | 22 (84%) | 23 (92%) | 23 (92%) |
| Urban (low or medium density) | 1 (4%) | 1 (4%) | 1 (4%) | 1 (4%) | 1 (4%) |

All continuous variables shown as median (interquartile range, IQR). [a]Data missing for one child in teduglutide group.

dose of budesonide; there were no clinical or laboratory consequences. One child receiving teduglutide was readmitted for vomiting to exclude intestinal obstruction (an Adverse Event of Special Interest) but the illness resolved uneventfully and was adjudicated as unrelated to the investigative product. Laboratory abnormalities occurred in 702 instances, including baseline abnormalities, but did not differ significantly between arms and were all adjudicated as unrelated to the investigative products (Supplementary Table S4).

## Discussion

Despite implementation of current treatment guidelines for complicated SAM, mortality remains unacceptably high, particularly in settings with high HIV burden. There is an urgent need for transformative approaches that modify underlying pathogenic pathways[39]. We identified intestinal mucosal damage as a promising target for intervention[27]. This multi-arm phase II clinical trial evaluated four potential new therapies to promote mucosal healing, which were each compared with standard care. Teduglutide showed benefit on the primary endpoint, a composite of three faecal inflammatory markers widely used to define enteropathy in malnourished children. Teduglutide is a GLP-2 agonist widely used in intestinal failure, a clinical syndrome with features in common with the more severe cases of malnutrition enteropathy. Our findings suggest that GLP-2 agonism similarly enhances mucosal healing in children with SAM. No other intervention significantly differed from standard care for the primary

outcome. However, budesonide, colostrum and N-acetyl glucosamine reduced the systemic inflammatory marker CRP, which is potentially clinically important as CRP is a predictor of mortality[40]. The increase in crypt depth observed with teduglutide and bovine colostrum likely indicates enhanced mucosal regeneration, as these agents both showed some reductions in inflammatory biomarkers (faecal composite score for teduglutide, CRP for colostrum). N-acetyl glucosamine reduced diarrhoea, which is independently associated with mortality in SAM, potentially reflecting restoration of glycocalyx composition[32] and/or inhibition of enteropathogen colonisation[41]. Collectively, our data identify teduglutide as the leading candidate for future trials, but also suggest there may be benefits from the other treatments, which all have distinct mechanisms of action. This trial highlights the importance of measuring multiple biomarkers, which capture different pathological domains of malnutrition enteropathy. A larger-scale trial of single or combined interventions with outcomes including mortality and readmission is now warranted.

Our hypothesis was that one or more trial interventions could aid mucosal healing, reducing enteropathy, inflammation, and microbial translocation. We selected three faecal biomarkers of mucosal damage and inflammation as a composite primary endpoint, on which we based our sample size calculations. However, we acknowledged a priori that our interpretation would draw on the full range of endpoints, since no single biomarker or composite score captures the complexity of pathogenesis linking mucosal damage to mortality[27].

**Table 3 | Primary and secondary endpoint final values following treatment**

| | Colostrum n = 25 | N-acetyl glucosamine n = 24 | Teduglutide n = 26 | Budesonide n = 24 | Standard care n = 23 |
|---|---|---|---|---|---|
| **Primary endpoint** | | | | | |
| Composite score[a] | 2.04 | 2.29 | 1.78 | 2.03 | 2.92 |
| Individual faecal biomarkers: | | | | | |
| Myeloperoxidase (µg/mL) | 0.67 (0.31–2.03) | 1.08 (0.55–1.96) | 0.59 (0.35–1.19) | 0.84 (0.48–1.2) | 1.06 (0.26–2.98) |
| Alpha-1 antitrypsin (mg/g) | 0.33 (0.18–0.79) | 0.30 (0.18–0.81) | 0.41 (0.16–0.60) | 0.39 (0.22–0.81) | 0.23 (0.16–0.78) |
| Neopterin (µmol/L) | 0.56 (0.28–0.89) | 0.76 (0.38–1.13) | 0.62 (0.44–0.99) | 0.53 (0.25–0.81) | 0.77 (0.33–1.07) |
| **Secondary endpoints** | | | | | |
| Plasma LBP (µg/mL) | 7.74 (4.21–8.90) | 5.73 (4.43–6.99) | 5.09 (4.00–6.45) | 5.87 (3.90–7.79) | 5.38 (4.40–9.88) |
| Plasma soluble CD14 (µg/mL) | 2.04 (1.82,2.31) | 2.17 (1.72,2.45) | 2.16 (1.69,2.53) | 1.87 (1.58,2.39) | 2.24 (1.58,2.96) |
| Plasma CD163 (ng/mL) | 1043 (767,1592) | 1239 (883,1610) | 1139 (920,1517) | 1029 (573,1409) | 1071 (896, 1880) |
| Plasma C-Reactive Protein (mg/L) | 1.13 (0.42–3.68) | 0.64 (0.24–5.60) | 1.03 (0.46–3.43) | 0.70 (0.31–1.55) | 2.08 (0.27–9.61) |
| Plasma intestinal fatty acid binding protein (iFABP) (pg/ml) | 1569 (1104–2494) | 1652 (909–3943) | 1989 (1371–2893) | 1656 (1208–3041) | 1104 (760–3925) |
| Plasma GLP-2 (ng/mL) | 5.24 (3.57–6.77) | 4.18 (2.97–7.18) | 4.86 (2.99–5.46) | 3.86 (2.62–4.91) | 4.22 (2.69–5.41) |
| Plasma IGFBP-3 (ng/mL) | 207 (63, 403) | 161 (99, 460) | 201 (144, 257) | 198 (86, 282) | 207 (91, 317) |
| Days with diarrhoea (days 1–15) | 7/358 days of observation | 4/338 | 18/373 | 6/338 | 18/332 |
| Days with fever (days 1–15) | 18/358 | 9/338 | 12/373 | 12/338 | 10/332 |
| Days with oedema (children with oedematous SAM only) (days 1–15) | 74/244 | 61/239 | 56/303 | 57/240 | 54/260 |
| Villus height (µm) | 210 (195, 213) | 215 (170, 258) | 231 (168, 263) | 203 (201, 246) | 193 (186, 213) |
| Crypt depth (µm) | 217 (202, 222) | 135 (117, 148) | 197 (149, 221) | 170 (161, 188) | 151 (136, 162) |
| Epithelial perimeter, representing surface area (µm per µm of length of muscularis mucosae) | 2.73 (2.57, 2.78) | 2.64 (2.22, 4.13) | 3.03 (2.39, 3.46) | 2.61 (2.28, 3.33) | 2.89 (2.47, 3.86) |

Data shown are median (IQR) for final (day 15) values. [a]Composite of faecal myeloperoxidase (MPO), neopterin (NEO), and alpha1-antitrypsin (AAT), calculated as 0.2xMPO + 2xAAT + NEO[23]. All statistical testing was 2-sided.

The effects of trial treatments on different secondary endpoints suggests that each may benefit specific domains of gut dysfunction in SAM[37].

The potential usefulness of teduglutide is limited by its expense and subcutaneous route of administration. However, many therapies introduced at high prices fall in cost once volume of use increases. By contrast budesonide, which reduced the inflammatory markers CRP and sCD163, is far less costly and easier to administer. Neither should be implemented as part of treatment for SAM without further trial evidence, but the TAME trial demonstrates that use is likely to be safe, and confirms mucosal healing as a promising strategy in severe malnutrition. Although colostrum did not affect the primary endpoint, it increased both plasma levels of GLP-2 (though only after log transformation) and crypt depth. Colostrum contains GLP-2 at around 5 ng/ml and supplementation with colostrum in juvenile pigs undergoing intestinal resection increased plasma GLP-2 levels[42].

As expected, adverse events were common but serious adverse events uncommon, and there were no events considered related to trial medications. The safety of long-term teduglutide has been assessed in intestinal failure and considered acceptable[35]. No Adverse Events of Special Interest were observed. A trial of mesalazine, another anti-inflammatory drug used in IBD, suggested safety in children with SAM[43], and our data extend these findings by showing that these medications are also safe. Mortality was low compared to our historical data[9,11]. This may be due to the enhanced medical and nursing care usually associated with conduct of a clinical trial, but probably also relates to our caution in focusing on inclusion of clinically stable children. Given our experience of high early mortality in children with complicated SAM, we adopted this strategy to reduce the likelihood of serious adverse events in this phase II trial. In future it would appear desirable to bring forward treatments to the day of admission, when

they might be of greatest potential benefit. Treatment duration was 14 days, representing the period of greatest mortality risk in hospitalised children; however, we and others have reported that unacceptably high mortality continues for 48 weeks following hospital discharge[2,11,13]. It is therefore possible that longer duration of therapies for mucosal healing could be of benefit.

We recognise several limitations. Due to restrictions on recruitment and difficulties transporting medicines and reagents during the COVID-19 pandemic, our enrolment was reduced. However, withdrawal and mortality were much lower than anticipated and we could therefore retain adequate power for the primary endpoint with a smaller sample size. Because this trial was conducted in hospital, adherence to medication was very high, with 98% of children completing intended therapy. This may be unattainable in real-world settings, but overall this trial demonstrated proof of concept for the therapies chosen. Our results were consistent across endpoint domains and biologically plausible for known mechanism of action of each agent, increasing confidence in our findings. Due to the short intervention duration, we saw limited impact on clinical outcomes such as growth. However, future trials powered for clinically important outcomes could explore the optimal timing, dosage and duration of intervention. There are also some challenges in interpreting the biomarkers used in this trial. Faecal biomarkers are subject to dilutional considerations, especially when a significant proportion of the trial participants have diarrhoea. It is also true that the biology of these biomarkers is not fully established. Markers such as myeloperoxidase and neopterin are elaborated by leukocytes, and reflect intestinal mucosal inflammation, but intestinal permeability and trafficking of leukocytes to the gut can be altered in the presence of systemic infections[44,45]. Neopterin is synthesised in response to interferon-γ and generally reflects Th1-mediated inflammation. α1-antitrypsin is usually

**Table 4 | Effect sizes in ANCOVA models[a] compared to standard care, with 90% CIs**

| | Colostrum n=25 | N-acetyl glucosamine n = 24 | Teduglutide n = 26 | Budesonide n = 24 |
|---|---|---|---|---|
| Composite score relative to Standard Care, adjusted for core covariates | −0.58 (−1.4, 0.23) P = 0.24 | −0.20 (−1.01, 0.60) P = 0.67 | −0.89 (−1.69, −0.10) P = 0.07 | −0.50 (−1.33, 0.33) P = 0.32 |
| Plasma intestinal fatty acid binding protein (IFABP) | −152 (−1528, 1223) P = 0.86 | 608 (−749, 1966) P = 0.46 | 170 (−1156, 1497) P = 0.83 | 28 (−1354, 1409) P = 0.97 |
| Plasma LBP (μg/mL) | −0.97 (−2.73, 0.79) P = 0.36 | −1.38 (−3.10, 3.53) P = 0.19 | −1.44 (−3.12, 0.24) P = 0.16 | −1.44 (−3.20, 0.32) P = 0.18 |
| Plasma soluble CD14 (μg/mL) | −0.10 (−0.46, 0.27) P = 0.65 | 0.26 (−0.11, 0.62) P = 0.25 | 0.19 (−0.16, 0.54) P = 0.38 | 0.08 (−0.29, 0.45) P = 0.73 |
| Plasma soluble CD163 (ng/mL) | −212 (−543, 119) P = 0.29 | −191 (−519, 138) P=0.34 | −320 (−642, 3.0) P=0.10 | −405 (−738, −73) P=0.05 |
| Plasma C-Reactive Protein (mg/L) | −5.9 (−10.3, −1.4) P = 0.03 | −4.8 (−9.2, −0.5) P = 0.07 | −3.0 (−7.2, 1.2) P = 0.24 | −5.2 (−9.6, −0.8) P = 0.05 |
| Plasma GLP-2 (ng/mL) | 1.4 (−0.2, 3.0) P = 0.14 | 0.92 (−0.6, 2.4) P = 0.31 | 0.69 (−0.8, 2.2) P = 0.44 | 0.007 (−1.5, 1.5) P = 0.99 |
| Days with diarrhoea incidence rate ratio (IRR) (90%CIs) of the mean days of duration relative to standard care Zero-inflated negative binomial[b] | 0.61 (0.29, 1.27) P = 0.27 | 0.11 (0.04, 0.33) P = 0.001 | 0.83 (0.42, 1.65) P = 0.66 | 0.96 (0.42, 2.21) P = 0.93 |
| Days with fever IRR (90%CIs) Negative binomial[c] | 2.35 (0.72, 7.69) P = 0.23 | 1.13 (0.40, 3.23) P = 0.85 | 1.64 (0.56, 4.81) P = 0.45 | 1.62 (0.57, 4.61) P = 0.45 |
| Days with oedema IRR (children with oedematous SAM only) (90%CIs) Negative binomial[d] | 1.27 (0.91, 1.78) P = 0.24 | 1.13 (0.73, 1.75) P = 0.63 | 1.20 (0.89, 1.62) P = 0.31 | 1.32 (0.93, 1.88) P = 0.19 |
| Change in weight (kg) between baseline and day 15 | 0.27 (−0.35, 0.58) P = 0.14 | 0.05 (−0.25.0.36) P = 0.77 | 0.24 (−0.06, 0.54) P = 0.19 | 0.26 (−0.05, 0.57) P = 0.17 |
| Change in height (cm) between baseline and day 15 | 0.15 (−0.28, 0.59) P = 0.55 | 0.06 (−0.38, 0.49) P = 0.83 | 0.16 (−0.26, 0.58) P = 0.53 | −0.04 (−0.48, 0.40) P = 0.89 |
| Change in MUAC (cm) between baseline and day 15 | 0.22 (−0.17, 0.60) P = 0.35 | 0.16 (−0.55, 0.22) P = 0.49 | 0.24 (−0.13, 0.62) P = 0.29 | 0.15 (−0.24, 0.55) P = 0.52 |

Effect sizes are shown (with 90%CIs) for ANCOVA treatment effects, or incidence rate ratio (IRR) for negative binomial; the precise models used for incidence rate data are indicated in italics. [a]Composite of faecal myeloperoxidase (MPO), neopterin (NEO), and alpha1-antitrypsin (AAT), calculated as 0.2×MPO + 2×AAT + NEO[33]. All statistical testing was 2-sided. [a]Adjusted for sex (male/female); oedema (yes/no); HIV status (yes/no); diarrhoea (yes/no); baseline biomarker value (continuous); and study site (Lusaka/Harare). MUAC, mid upper arm circumference. [b]Adjusted for sex (male/female) and HIV status (yes/no). [a]Adjusted for sex (male/female); HIV status (yes/no); baseline WLZ scores (continuous); and study site. [d]Adjusted for sex (male/female); baseline severity of oedema; HIV status (yes/no); baseline WLZ scores (continuous); and study site.

considered a biomarker of protein loss into the gut, but transcriptomic data reveal that it is expressed in the mucosa[26,46]. These considerations need to be taken into account in future work. We have no ready explanation for the heterogeneity between study sites. We have previously noted mortality differences between Zambia and Zimbabwe, and there are minor differences in protocol implementation (such as when children are ready for discharge) which might explain some of these effects. This heterogeneity also needs to be taken into account in future work as it underlines the value of performing trials in different countries.

Our findings demonstrate a biologically plausible new treatment paradigm for children with complicated SAM. Intestinal damage is ubiquitous in children with SAM, driving systemic inflammation, contributing to stunting and developmental impairment and increasing mortality. No interventions for malnutrition enteropathy are currently available. We have shown that a short course of therapy added to standard care, aimed at restoring mucosal integrity, can ameliorate underlying pathogenic pathways. Further trials should evaluate these interventions for their effects on mortality, clinical recovery and long-term nutritional restoration to improve the outcomes of children with complicated SAM. Combinations of interventions would be interesting to evaluate in future trials since their distinct mechanisms of action and potential to target multiple domains of malnutrition enteropathy concurrently, may plausibly lead to greater mucosal healing and clinical recovery through synergistic effects.

## Methods
Ethics approval was obtained from the University of Zambia Biomedical Research Ethics Committee (006-09-17), the National Health Research Committee of Zambia, the Zambia Medicines Regulatory Authority (CT 082/18), the Joint Research Ethics Committee of Harare Central Hospital (JREC/66/19), the Medicines Control Authority of Zimbabwe (CT/176/2019), and the Medical Research Council of Zimbabwe (MRCZ/A/2458). The trial was conducted in accordance with the principles of the Declaration of Helsinki. The trial was monitored by a Data Safety and Monitoring Board. The trial was registered at www.clinicaltrials.gov (NCT03716115) and first posted on 23rd October 2018, prior to patient enrolment, and the protocol was published[27]. A CONSORT checklist containing information reporting a randomized controlled trial has been included in Supplementary Note 1. The trial protocol and statistical analysis plan are included in Supplementary Notes 2 and 3.

### Recruitment, inclusion and exclusion criteria
Children were recruited from 4th May 2020 to 27th April 2021, and the trial closed on 25th May 2021 after the completion of the period of follow up of the last child. Potentially eligible children were identified by the study nursing teams, and written permission to screen was obtained from caregivers. A detailed information sheet in local languages was discussed with caregivers prior to seeking consent. Weight, length and mid-upper arm circumference (MUAC) were measured three times, and eligibility ascertained. Children of either sex were eligible for inclusion if aged 6-59 months and hospitalised with SAM (using WHO definitions: WLZ <−3, and/or MUAC <11.5cm, and/or bilateral pedal oedema), with written, informed consent from the primary caregiver. Children were excluded if unstable (shocked, hypothermic, hypoglycaemic, impaired consciousness), under 5kg body weight, had conditions impairing feeding, haemoglobin below 6 g/dl, or if their caregiver would not consent to child HIV testing or to remaining in hospital throughout the treatment course. Additional exclusion criteria were contraindications to any treatment or other factors that might prejudice study completion or analysis. No payments were made for participation, but transport refunds were made on discharge and on review to permit return home using public transport.

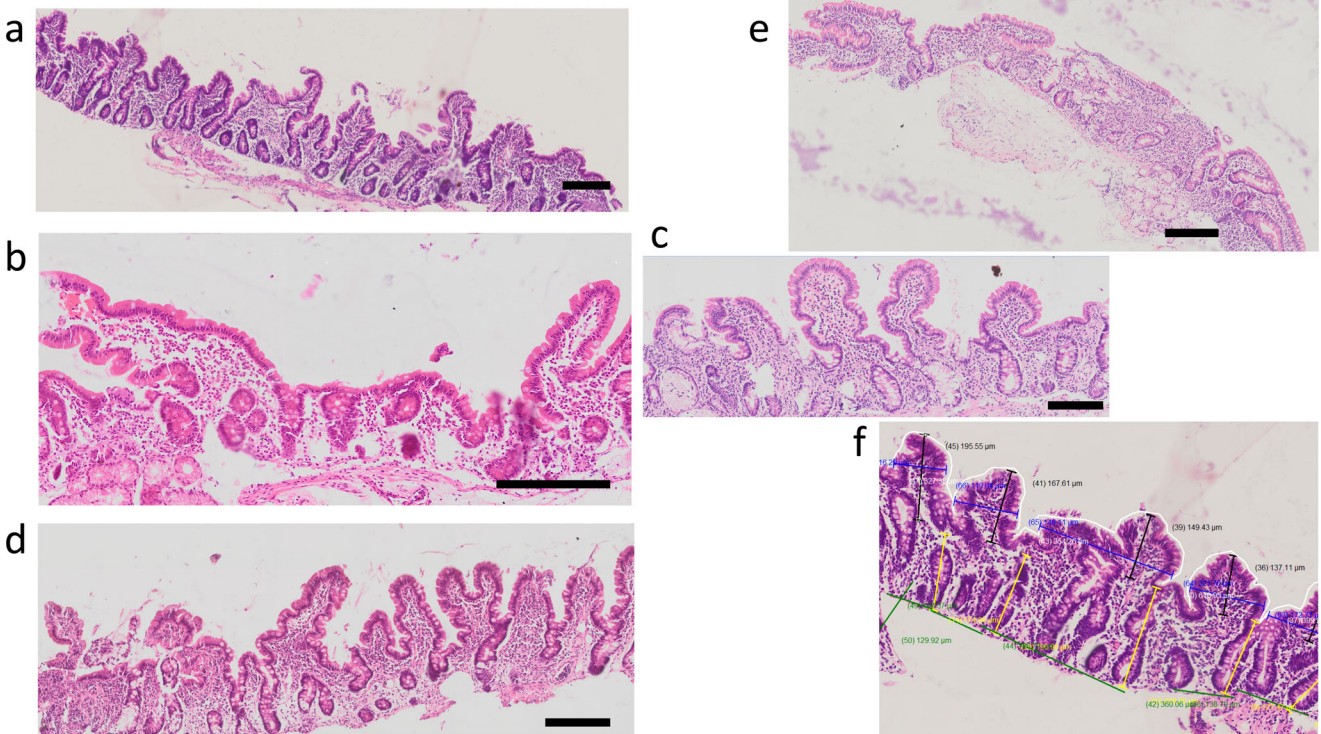

**Fig. 2 | Biopsy images from children after 14 days of treatment.** Biopsies are from children treated with **a** colostrum, **b** N-acetyl glucosamine, **c** teduglutide, **d** budesonide, and **e** standard care. Morphometric analysis is shown in panel **f**. Scale bars show 200 μm. These biopsies were selected from 22 biopsies from 25 children: 6 in the teduglutide group, 5 in the colostrum group, 5 in the budesonide group, 5 in the standard care group, and 4 in the N-acetyl glucosamine group. Individual data from morphometric analysis are shown in Fig. 3.

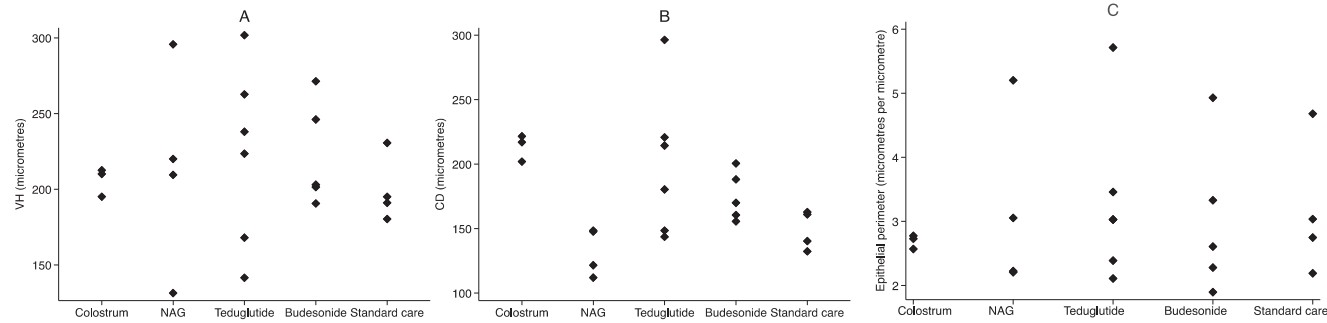

**Fig. 3 | Mucosal morphometry.** Measurements of villus height (VH) and crypt depth (CD) in 22 biopsies with satisfactory orientation obtained from children completing 14 days of treatment. a villus height ($P$ = 0.84 by Kruskal-Wallis test across all groups). **b** crypt depth ($P$ = 0.02 by Kruskal-Wallis test; by Dunn's test $P$ = 0.01 for colostrum and $P$ = 0.049 for teduglutide). c, epithelial surface area. NAG, N-acetyl glucosamine. Source Data are provided as a Source Data File (Dataset 1).

## Trial procedures

Children were randomised equally to each of the four interventions, or standard care. Trial identification numbers were allocated sequentially in each site, with randomisation carried out by opening a sealed envelope bearing the corresponding number. The randomisation sequence was prepared by the trial statistician (KVB), stratified by study site, in random permuted blocks of variable size. Interventions began as soon as possible after baseline samples were collected. Children were managed by a team of nurses providing 24-hour cover, ensuring all treatments were administered and adverse events recorded. Study doctors (KC, GT) reported clinical progress daily using a standardised form. All treatment courses were 14 days. Samples for endpoint analysis were collected on day 15 (permissible window 15-19); children were then discharged if ready and reviewed on day 28 (window 28-42). Blood samples for safety monitoring (full blood count, renal and liver function, phosphate) were collected at baseline, 5 and

15 days post-randomisation. We enrolled children with predominantly oedematous SAM, and a high prevalence of HIV infection, and were not powered to evaluate effects in different subgroups of children. Tuberculosis was diagnosed clinically, and specifically searched for in any child with respiratory symptoms, using microscopy and culture of nasogastric aspirates whenever possible, chest X-ray and urine lipoarabinomannan according to clinical protocols.

## Investigational products

Bovine colostrum (supplied by Colostrum UK) and N-acetyl glucosamine (Blackburn Distributions, UK) are nutraceuticals, regarded as safe and not licensed as medicines. They were provided as powder and encapsulated to ensure accurate dosing (colostrum 1.5 g 8-hourly throughout; N-acetyl glucosamine 300 mg 8-hourly days 1–7, 600 mg 8-hourly days 8–14). The dose of colostrum (4.5 g/day orally or via nasogastric tube) was chosen to be similar to those used in other

published studies involving children. Ismail et al. treated premature infants with a dose of approx. 0.5–1 g colostrum/kg/day to examine gut immune priming[47], and Barakat & Omar used 3 g/day of colostrum for children aged 6 months to 2 years suffering from acute diarrhoea[48]. The dose of N-acetyl glucosamine was based on our previous study in paediatric Crohn's disease, where daily dosage of 6 grams augmented expression of epithelial glycosaminoglycans without evidence of adverse effects[33]. Intravenous doses of up to 100 mg have been tolerated in adults without adverse effects[33] and breastfed newborns consume 650–1500 mg of n-acetyl glucosamine per litre of human breast milk from well-nourished mothers[49]. Budesonide 0.5 mg and 1 mg respules (Alliance Healthcare, UK) are designed for nebuliser therapy but used off-licence orally for gastrointestinal therapy. These were opened on the ward and administered orally or by nasogastric tube. Dosage was 1 mg 8-hourly during days 1–7, 1 mg 12-hourly during days 8–11, and then 0.5 mg 12-hourly during days 12–14. The dosage was derived from studies in paediatric Crohn's disease, where 9 mg enteric-coated daily budesonide proved equally efficacious to 40 mg prednisolone but with substantially reduced adverse effects[50]. Teduglutide (Revestive, Takeda) is licensed widely for intestinal failure but never previously evaluated in SAM. It was provided as 1.25 mg vials which were stored at 4–8 °C, and given by subcutaneous injection (0.05 mg/kg daily, based on weight measured on days 1 and 8). Prior stability testing carried out by Takeda confirmed that the opened vial is stable at 4–8 °C for 24 hours so each vial provided two doses, drawn 24 hours apart. Site rotation was marked on a map of anatomical sites as recommended by the manufacturer. The selected dose for teduglutide (0.05 mg/kg) was found to be the most effective dose in a phase 3, 12-week paediatric trial when compared to two lower doses of 0.025 mg/kg and 0.0125 mg/kg[51]. It is the dose currently approved by the FDA in the US and the EMA in Europe.

### Evaluation of endpoints

The primary endpoint was the day 15 composite faecal inflammatory biomarker score, comprising myeloperoxidase, neopterin and $\alpha_1$-antitrypsin[27]. Secondary endpoints at day 15 were: changes in anthropometry; plasma biomarkers of enteropathy, microbial translocation and systemic inflammation (iFABP, LBP, CRP, sCD14, CD163, IGFBP-3, and GLP-2); days with diarrhoea, fever, and oedema; and adverse events. For children who were potty trained, stool was collected using a clean pot and then the required amount was transferred to a sterile stool container using a scoop. For those who are were not potty trained, diapers were used. The diapers were put inside out so that the plastic layer was next to the skin. A scoop was used to place a sample in a sterile stool container. The collected sample was then put in a cooler box with ice packs immediately. A sample transmittal form was used to keep track of the transit time from point of collection to receipt in the laboratory. Nursing staff stayed in communication with lab staff to ensure rapid delivery of samples to the laboratory.

Biomarkers were assayed by ELISA (Supplementary Table S5) by laboratory scientists (KZ, KM, EB) blinded to study arm, and recalculated independently (by RN and JS) from raw data on harmonised Gen5 software (Biotek/Agilent, Santa Clara, CA). Serum albumin and lipopolysaccharide, though pre-specified as endpoints, were not included due to failing quality control checks leading to low concordance between sites. IGF-1 values were very close to zero; as insufficient plasma was available for re-testing these data have been omitted. Lactulose/rhamnose urinary excretion tests were only performed on children undergoing endoscopy and only 14 valid data pairs were obtained; these data are therefore not shown.

### Endoscopy

A subgroup of children in Lusaka additionally underwent endoscopy for duodenal biopsy between days 15 and 19; only the Lusaka site was selected for this due to its considerable experience in paediatric endoscopy over many years. Except for two periods when endoscopy instruments required repair, children were selected sequentially, provided there were no haematological or anaesthetic contraindications. Sedation was administered by an anaesthetist (HS or MZ) and biopsies were collected from the second part of the duodenum using a Pentax 2490i paediatric gastroscope. Biopsies were orientated under a dissecting microscope and fixed before processing into paraffin blocks, sectioning and staining. Slides were scanned at 20x magnification on an Olympus VS-120 scanning microscope and blinded morphometry was performed by a single observer (CM, confirmed by PK) on all villus and crypt units identifiable in well-orientated parts of sections of each biopsy. The criterion used for suitable orientation was that crypts should be visualised throughout their length (see Fig. 2, and reference[22]), and then the boundary between crypt and villus compartments delineated. Crypt depth was measured in micrometres (μm) from this boundary to the furthest point of the base of the crypt, where the basement membrane would be expected. Villus height was measured in μm from the boundary to the villus tip in a straight line. Epithelial surface area was measured as the perimeter of the villi where muscularis mucosae could be measured, and expressed per micrometre of muscularis mucosae. Any portions of these sections where crypts were not visualised along their entire length were deemed poorly orientated and not used for morphometry. The median number of villi measured was 6 (IQR 4-9; range 3-13).

### Adverse events

Adverse events between enrolment and day 15 or day 28 were reported in real time and reviewed for seriousness, severity, relatedness and expectedness; all serious adverse events were reported to the Sponsor (Queen Mary University of London), ethics committees and national trial regulators. Severity was categorised as mild, moderate or severe using the DAIDS classification (https://rsc.niaid.nih.gov/clinical-research-sites/daids-adverse-event-grading-tables), and all AEs reported monthly to the Data Monitoring and Ethics Committee (DMEC). Three Adverse Events of Special Interest (AESIs) were specifically sought: intestinal obstruction or volume overload for teduglutide, and osmotic diarrhoea for colostrum and N-acetyl glucosamine. Haematology and biochemistry results at baseline, days 5 and 15 were graded using DAIDS tables.

### Sample size

The planned sample size was 225 children (45 in each arm), based on the composite biomarker score[27]. Enrolment was slowed by COVID-19, but trial losses were much lower than anticipated (3% observed versus 20% anticipated). The Trial Steering Committee and DMEC therefore reviewed the sample size in January 2021 once 82 children had been enrolled. The decision was made to reduce the sample size, based on a Cohen's d effect size of 0.3, with 80% power and 90% confidence, and conservative correlation between baseline and follow-up estimates of 0.5, requiring 23 per group across 5 groups to analyse the primary outcome by ANCOVA. Allowing for 5% losses, the sample size of 115 was rounded up to 125 participants in total (25 per group).

### Statistical analysis

Per protocol analyses were pre-specified[27], and all hypothesis testing was 2-sided. Statistical analysis was performed in Stata 17 (Stata corp, College Station, Texas). Analysis of primary and secondary endpoints was based on comparison against standard care. ANCOVA was used to model final endpoint values, with adjustment for core baseline value, sex, baseline presence of oedema, HIV status, baseline diarrhoea, baseline WLZ score, and study site. Mortality was low (3 deaths) so could not be analysed statistically. Covariates chosen were pre-specified to take into account important elements of pathophysiology (worse outcome in HIV infection and oedematous malnutrition and in children with diarrhoea[52]) and to allow for possible differences between the two countries. For some secondary endpoints negative

binomial models were constructed which used a smaller set of adjustment variables (sex & HIV) due to model constraints (Table 2). Anthropometric measurements were calculated as change from baseline. The endoscopy subset was analysed separately, comparing post-treatment morphometric measurements by Kruskal-Wallis test, followed by Dunn's test. Treatment effects were deemed statistically significant if the $P$ value was <0.1 when compared to the control arm, as pre-specified. No adjustments of the false-positive (type I) error rate were planned, in line with the general consensus that adjustment for type I error rate is not required in exploratory multi-arm multi-stage trials in Phase II within the treatment development framework[27,53].

## Reporting summary

Further information on research design is available in the Nature Portfolio Reporting Summary linked to this article.

## Data availability

Data supporting the findings of this study are available in the article and in its Supplementary Information. Source data are provided as a Source Data file for Fig. 3, and have also been deposited in Figshare under accession code https://doi.org/10.6084/m9.figshare.24442699. The data uploaded to figshare include deidentified individual participant data, trial protocol, and statistical analysis plan.

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

## Acknowledgements

We are grateful to the following nurses from the wards and endoscopy unit of UTH: Evelyn Nyendwa, Esther Chilala, Andreck Tembo, Lucy Macwani, Dalitso Tembo, Mary Phiri, Elaine Brittel Sikuyuba, Sophreen Mwaba, Gwendolyn Nayame, Joyce Sibwani, Rose Soko, Kashinga Maseko, and Mulima Mwiinga. We are grateful to the nursing team in Harare Central Hospital: Sarudzai Murumbi, Tariro Zure, and Lucia Manyatera. We also sincerely thank Mr Rizvan Batha of Barts Health Pharmacy for assistance with procurement of investigational products. We are very grateful to the Trial Steering committee (Professors Jay Berkley, Ian Sanderson, and James Wason) and Data Monitoring and Ethics Committee (Professor Jim Todd, Doctors Rose Kambarami, Veronica Mulenga, and Philip Ayieko). The TAME trial was sponsored by Queen Mary University of London, but the Sponsor played no role in study design, data collection, analysis, or manuscript writing. The trial was funded by a grant from the Medical Research Council (UK), number MR/P024033/1. AJP and JPS are funded by Wellcome (108065/Z/15/Z for AJP, and 220566/Z/20/Z for JPS). Takeda UK provided teduglutide at a discounted price.

## Author contributions

KC, MBD, BA, SHM, RJP, SH, AJP and PK initiated and designed the trial. KC, GT, DN, AD, NC, RM, LK, BM and LL designed the trial instruments and data collection procedures. KC, MBD, BA, GT, DN, AD, NC, and RM carried out the daily clinical care and data collection. KZ, KM, CM, EB and VM were responsible for laboratory operating procedures, data acquisition and processing. LK, BM and LL undertook data entry and cleaning. MC, VS and LL undertook monitoring and quality control of the trial. BS and SM designed and implemented the pharmacy and pharmacovigilance procedures. MZ and HS undertook anaesthetic procedures and ensured the safety of children undergoing endoscopy. Analysis was carried out by KVB, JPS, LL, AJP and PK; AJP and PK wrote the initial draft which was revised by all authors.

## Funding

The Medical Research Council (UK) funded the study. Takeda UK provided tedu-glutide at a discounted price.

## Competing interests

RJP was previously an external consultant to Colostrum UK which provided the bovine colostrum used in these studies. RJP has also been an external consultant to Sterling Technology (USA) and an employee of Pantheryx Inc (USA) who produce and distribute bovine colostrum. There was no bovine colostrum company involvement in the production of this article or editing of its content. SH has had funding for teduglutide studies and lectured and participated in advisory boards on behalf of Takeda. The remaining authors declare no competing interests.

## Additional information

Kanta Chandwe[1,10], Mutsa Bwakura-Dangarembizi[2,3,10], Beatrice Amadi[1], Gertrude Tawodzera[2], Deophine Ngosa[1], Anesu Dzikiti[2], Nivea Chulu[1], Robert Makuyana[2], Kanekwa Zyambo[1], Kuda Mutasa[2], Chola Mulenga[1], Ellen Besa[1], Jonathan P. Sturgeon[2,4], Shepherd Mudzingwa[2], Bwalya Simunyola[1], Lydia Kazhila[1], Masuzyo Zyambo[5], Hazel Sonkwe[5], Batsirai Mutasa[2], Miyoba Chipunza[1], Virginia Sauramba[2], Lisa Langhaug[2], Victor Mudenda[1], Simon H. Murch[6], Susan Hill[7], Raymond J. Playford[8,9], Kelley VanBuskirk[1], Andrew J. Prendergast[2,4] & Paul Kelly[1,4] ✉

[1]Tropical Gastroenterology & Nutrition group, University of Zambia School of Medicine, Nationalist Road, Lusaka, Zambia. [2]Zvitambo Institute for Maternal and Child Health Research, McLaughlin Avenue, Meyrick Park, Harare, Zimbabwe. [3]Faculty of Medicine and Health Sciences, University of Zimbabwe, Parirenyatwa Hospital, Harare, Zimbabwe. [4]Blizard Institute, Queen Mary University of London, Newark Street, London, UK. [5]Department of Anaesthesia, University of Zambia School of Medicine, Nationalist Road, Lusaka, Zambia. [6]Warwick University Medical School, Coventry, UK. [7]Great Ormond Street Hospital, London, UK. [8]University of West London, Ealing, London, UK. [9]University College Cork, College Road, Cork, Ireland. [10]These authors contributed equally: Kanta Chandwe, Mutsa Bwakura-Dangarembizi. ✉e-mail: m.p.kelly@qmul.ac.uk

