## [Peer Review File · Nature Communications]

REVIEWER COMMENTS

Reviewer #1 (Remarks to the Author):

Review of NCOMMS-23-17948

The purpose of this multi-arm, phase II trial was to assess the safety and potential efficacy of four treatments on malnutrition enteropathy in children with severe acute malnutrition. Investigators found that teduglutide significantly reduced the primary composite endpoint compared to the standard care, while colostrum and budesonide showed promise on key secondary endpoints, though further trials are needed to establish efficacy on long term outcomes before treatment recommendations can be made. The design and analysis of this trial are sound but require additional details to clarify analytic decisions and interpretations of results, and to make the study reproducible. Many of these comments are minor but their inclusion are intended to make statistical and analytical decisions and interpretations of results clear to non-statistical experts. The comments are listed first in order of relative importance and next by location in manuscript.

1. Lines 177-179: “The increase in crypt depth observed with teduglutide and bovine colostrum likely indicates enhanced mucosal regeneration, as all agents reduced inflammatory biomarkers.” It is unclear which inflammatory biomarkers the authors are referring to in the latter portion of this statement. Only teduglutide significantly reduced the inflammatory markers making up the composite primary endpoint, though all other treatment groups had non-statistically significant ANCOVA point estimates that were lower than the ANCOVA point estimate for the standard of care group. Similar confusion is found on lines 210-212 (“By contrast budesonide, which had the greatest effect on inflammatory markers, is far less costly and easier to administer”), as budesonide was only statistically significant on the plasma CRP and sCD163 secondary endpoints.

2. The authors note in the Discussion that colostrum increased plasma GLP-2 and crypt depth. The former result is only statistically significant when plasma GLP-2 is log-transformed (no point estimate is provided), and this should be noted.

3. As a whole, the authors should more clearly delineate the unique goals of the primary and secondary endpoints either earlier in the manuscript or in the Methods section. While it is clearer in the abstract, it is slightly murky when reading the main text which aspects or domains of malnutrition enteropathy recovery the primary and secondary endpoints specifically represent, and what it means for a treatment to statistically significantly differ from standard care on the primary endpoint versus a secondary endpoint (and vice versa).

4. In the Methods section, the authors list the covariates by which they adjust their ANCOVA model for the primary results. I believe a sentence of two on why those covariates were included would improve others' understanding of the analysis.

5. In the Discussion, authors should elaborate on potential reasons why teduglutide may be efficacious on the primary composite endpoint but on none of the secondary endpoints (except crypt depth), and the implications (if they exist) of such discrepancy in future trials or potential treatment plans. This may be potentially resolved through the elaborations requested on comment 3.

6. The authors note in the text that endoscopy was only performed at the Lusaka site and note in the protocol that this was due to safety concerns for this procedure at the Harare site; this reason should be briefly mentioned in the main text to make it clear why the endoscopy subset only consists of Lusaka children.

7. The authors make it clear throughout the main text that the study under consideration is a phase II trial intended to explore the safety and ability of four treatments to improve malnutrition enteropathy outcomes in children with severe acute malnutrition, and that the results should not indicate recommendation of treatment but should advise future trials. The phase II nature of the trial should be made clear in either the abstract or title as well.

8. I personally find Supplementary Table 1 extremely confusing. It is unclear whether the site-stratified analyses were also performed under transformed endpoints, as their results appear in the same line as transformed analyses performed on the "whole group". If they were also transformed, it would stand to reason that non-transformed site-stratified analyses also be conducted and presented. There are also many endpoints that were not transformed but appear in the table with transformation listed as "not done". I would suggest first creating separate tables for reporting results of transformed-endpoint analyses performed on the whole group and the non-transformed site-stratified analyses. Additionally, I would remove endpoints from the "transformed analysis" table that did not undergo this additional exploratory/sensitivity analysis; they may be listed in a table caption or footnote to acknowledge transformations on those endpoints were not explored. Finally, the authors state on lines 128-129 of the main text that there were notable site-specific differences in CRP, sCD14, sCD163, and iFABP but do not elaborate on this point either in the Results or Discussion; the authors should explain what these differences are and the implications of site differences on these endpoints to the overall interpretation of study results.

9. There are several portions of the manuscript where acronyms are used but are not introduced until later in the text. For example, many acronyms such as WLZ score are introduced in the Methods section but, as the section itself does not appear until after the Discussion, discussion of WLZ scores prior to the Methods is slightly confusing. Also see lines 93 and 95 where the acronym IBD is spelled out on line 95

but first used on line 93. It would be appreciated if the authors reviewed the manuscript ensure that all acronyms are fully spelled out upon first use, including the acronym for the name of the trial.

10. There appears to be inconsistent formatting of how confidence intervals are reported throughout the manuscript (sometimes in parentheses or square brackets, sometimes not). Either format is appropriate but should be applied consistently.

11. The authors sometimes list the IQR for a measure in parentheses, but it is not always clear that is what is being reported unless the reader is directly referencing the accompanying table. For clarity, please ensure that all statistics and summary measures are labeled in text, regardless of whether they are additionally labeled in subsequent tables or figures.

12. The authors should mention in the Methods section whether statistical tests were conducted as one- or two-sided.

13. An explicit hypothesis should be stated toward the end of the Introduction section.

Reviewer #2 (Remarks to the Author):

This is an outstanding report of a Phase II clinical trial of four novel compounds designed to decrease the burden of malnutrition enteropathy from a research team that has truly been the world leaders in this domain and have brought an impressive rigor and design to this relatively small clinical trial. Despite looking hard for things to critique, I have only relatively minor comments:

Table 1 – The tuberculosis data is a bit odd – how were these diagnosed? How did one child’s TB resolve in just the few days between admission and randomization? Was this simply because the TB was a presumptive clinical diagnosis and this was re-evaluated during the intervening time period? The edema numbers also shifted a bit; presumably this was due to their resolution on F-75?

Line 113 and Table 2 – Why was the primary endpoint this composite enteropathy score, rather than a *change* in composite score from the time of enrollment in the study? (Or more ideally from the time of admission to hospital, as above?)

Lines 115-116 and Table 2 – It would be fine to express that there is a trend towards improvement in the teduglutide group, but a p value of 0.07 doesn't quite meet traditional definitions of statistical significance.

Line 119 – Same regarding the p = 0.09 value here. (Yes, I question the utility of p values too.)

Line 127 – Same.

Reviewer #3 (Remarks to the Author):

The study described in the manuscript by a research group from Zambia and Zimbabwe aimed to test the effect of four intestinal health targeting interventions in hospitalized children with severe malnutrition. These interventions consisted of bovine colostrum, N-acetyl glucosamine, budesonide and teduglutide and were tested using a phase II trial. Enteropathy is common in children with malnutrition and likely contributes to the high mortality rate in this population. There are no current interventions to facilitate restoration of intestinal health in malnourished children. Therefore, the trial is timely and of significant global health importance. The different interventions are well justified and the trial design is mostly appropriate. However, I do have a number of concerns which I have listed below.

A comment comment relates to the primary outcome. In the introduction, it is stated a broad range of outcomes were selected, but the primary outcome is based on three biochemical parameters in stool which is confusing and also not consistent with the BMJ protocol article (2019). In addition, the rationale for the stool composite score and the specific stool proteins selected should be better explained and their limitations discussed in detail. Limitations include falsely high levels due to upper respiratory infections leading to ingestion of neutrophils, dilution of stool samples due to diarrhea. In general, there is still uncertainty how well the stool biochemical readouts reflect severity of intestinal disease and this should be more explicitly discussed. This also relates to the validity of the composite score.

Adding a combination arm to this trial would have had added value as the interventions target different enteropathy pathways and there is a reasonable chance for synergistic effects. The lack of a combination arm should be discussed as a limitation.

There are important methodological details missing:

-Details on stool collection, sample transportation and storage.

- Specifics on the ELISA's performed,
- how the morphometrics was performed including number of sections or villi visualized.
- rationale for specific dosing of the study drugs should be added.
- IGFBP-3 seems to be an outcome but not described in the methods

There are some inconsistencies between the published protocol manuscript in BMJ (2019) and the current manuscript. IGF-1 was listed as a secondary outcome, some of the morphometric data mentioned in the protocol paper is not presented in the current manuscript. This data needs to be added or if not possible, the discrepancy needs to be clarified.

The site specific differences in the primary outcome need to be part of the discussion as this could have implications for the generalizability. Can the authors speculate why children in these sites might respond differently and why there was no clear effect of any of the interventions on the primary outcome in the Lusaka site?

Line 119 mentions a site specific effect presented in Suppl Table 2, but appears to be an error.

Reviewer #1 (Remarks to the Author):

The purpose of this multi-arm, phase II trial was to assess the safety and potential efficacy of four treatments on malnutrition enteropathy in children with severe acute malnutrition. Investigators found that teduglutide significantly reduced the primary composite endpoint compared to the standard care, while colostrum and budesonide showed promise on key secondary endpoints, though further trials are needed to establish efficacy on long term outcomes before treatment recommendations can be made. The design and analysis of this trial are sound but require additional details to clarify analytic decisions and interpretations of results, and to make the study reproducible. Many of these comments are minor but their inclusion are intended to make statistical and analytical decisions and interpretations of results clear to non-statistical experts. The comments are listed first in order of relative importance and next by location in manuscript.

1. Lines 177-179: “The increase in crypt depth observed with teduglutide and bovine colostrum likely indicates enhanced mucosal regeneration, as all agents reduced inflammatory biomarkers.” It is unclear which inflammatory biomarkers the authors are referring to in the latter portion of this statement. Only teduglutide significantly reduced the inflammatory markers making up the composite primary endpoint, though all other treatment groups had non-statistically significant ANCOVA point estimates that were lower than the ANCOVA point estimate for the standard of care group. Similar confusion is found on lines 210-212 (“By contrast budesonide, which had the greatest effect on inflammatory markers, is far less costly and easier to administer”), as budesonide was only statistically significant on the plasma CRP and sCD163 secondary endpoints.

Thank you for pointing out this ambiguity. The effect of teduglutide on inflammatory biomarkers was principally noted on faecal myeloperoxidase, and narrowly missed significance for a reduction in sCD163. Budesonide had a significant effect on two biomarkers of systemic inflammation (CRP and sCD163). This has been clarified in the text (lines 177-181, 211-213).

2. The authors note in the Discussion that colostrum increased plasma GLP-2 and crypt depth. The former result is only statistically significant when plasma GLP-2 is log-transformed (no point estimate is provided), and this should be noted.

This has been amended in the text (lines 215-217).

3. As a whole, the authors should more clearly delineate the unique goals of the primary and secondary endpoints either earlier in the manuscript or in the Methods section. While it is clearer in the abstract, it is slightly murky when reading the main text which aspects or domains of malnutrition enteropathy recovery the primary and secondary endpoints specifically represent, and what it means for a treatment to statistically significantly differ from standard care on the primary endpoint versus a secondary endpoint (and vice versa).

This important point has been added to the Introduction (lines 96-102).

4. In the Methods section, the authors list the covariates by which they adjust their ANCOVA model for the primary results. I believe a sentence of two on why those covariates were included would improve others' understanding of the analysis.

This has been included in the Methods (lines 351-354).

5. In the Discussion, authors should elaborate on potential reasons why teduglutide may be efficacious on the primary composite endpoint but on none of the secondary endpoints (except crypt depth), and the implications (if they exist) of such discrepancy in future trials or potential treatment plans. This may be potentially resolved through the elaborations requested on comment 3.

This is an important point which we believe reflects the impact of different interventions on different domains of pathophysiology: budesonide on inflammation, N-acetyl glucosamine on the epithelial barrier, while teduglutide and colostrum might be expected to mediate epithelial restitution. This has been added to the Discussion (lines 190-192).

6. The authors note in the text that endoscopy was only performed at the Lusaka site and note in the protocol that this was due to safety concerns for this procedure at the Harare site; this reason should be briefly mentioned in the main text to make it clear why the endoscopy subset only consists of Lusaka children.

Endoscopy in children for research purposes is subject to considerable ethical and safety concerns. The Lusaka endoscopy unit has considerable experience of these procedures and the local team was comfortable with undertaking them. The Harare team does not have this experience and we felt this was not the time to introduce them. This has been introduced as advised (lines 312-322).

7. The authors make it clear throughout the main text that the study under consideration is a phase II trial intended to explore the safety and ability of four treatments to improve malnutrition enteropathy outcomes in children with severe acute malnutrition, and that the results should not indicate recommendation of treatment but should advise future trials. The phase II nature of the trial should be made clear in either the abstract or title as well.

This has been done (title and abstract, lines 42-43).

8. I personally find Supplementary Table 1 extremely confusing. It is unclear whether the site-stratified analyses were also performed under transformed endpoints, as their results appear in the same line as transformed analyses performed on the "whole group". If they were also transformed, it would stand to reason that non-transformed site-stratified analyses also be conducted and presented. There are also many endpoints that were not transformed but appear in the table with transformation listed as "not done". I would suggest first creating separate tables for reporting results of transformed-endpoint analyses performed on the whole group and the non-transformed site-stratified analyses. Additionally, I would remove endpoints from the "transformed analysis" table that did not undergo this additional exploratory/sensitivity analysis; they may be listed in a table caption or footnote to acknowledge transformations on those endpoints were not explored. Finally, the authors state on lines 128-129 of the main text that there were notable site-specific differences in CRP, sCD14, sCD163, and iFABP but do not elaborate on this point either in the Results or Discussion; the authors should explain what these differences are and the implications of site differences on these endpoints to the overall interpretation of study results.

This point is well made and this table has been completely re-worked.

9. There are several portions of the manuscript where acronyms are used but are not introduced until later in the text. For example, many acronyms such as WLZ score are introduced in the Methods section but, as the section itself does not appear until after the Discussion, discussion of WLZ scores prior to the Methods is slightly confusing. Also see lines 93 and 95 where the acronym IBD is spelled out on line 95 but first used on line 93. It would be appreciated if the authors reviewed the manuscript ensure that all acronyms are fully spelled out upon first use, including the acronym for the name of the trial.

This has been corrected (lines 92,127 and 88).

10. There appears to be inconsistent formatting of how confidence intervals are reported throughout the manuscript

(sometimes in parentheses or square brackets, sometimes not). Either format is appropriate but should be applied consistently.

These inconsistencies have been corrected (lines 124, and 133).

11. The authors sometimes list the IQR for a measure in parentheses, but it is not always clear that is what is being reported unless the reader is directly referencing the accompanying table. For clarity, please ensure that all statistics and summary measures are labeled in text, regardless of whether they are additionally labeled in subsequent tables or figures.

Agreed. This has been done (line 150).

12. The authors should mention in the Methods section whether statistical tests were conducted as one- or two-sided. All tests were two-sided. This has been included (line 352).

13. An explicit hypothesis should be stated toward the end of the Introduction section. This has been included (lines 88-90).

Reviewer #2 (Remarks to the Author):

This is an outstanding report of a Phase II clinical trial of four novel compounds designed to decrease the burden of malnutrition enteropathy from a research team that has truly been the world leaders in this domain and have brought an impressive rigor and design to this relatively small clinical trial.

Thank you

Despite looking hard for things to critique, I have only relatively minor comments:

Table 1 – The tuberculosis data is a bit odd – how were these diagnosed? How did one child's TB resolve in just the few days between admission and randomization? Was this simply because the TB was a presumptive clinical diagnosis and this was re-evaluated during the intervening time period? The edema numbers also shifted a bit; presumably this was due to their resolution on F-75?

We fully agree that the diagnosis of tuberculosis is very difficult in SAM. The diagnosis was often purely clinical, usually based on culture of nasogastric aspirates or on clinical suspicion. The numbers have been checked and we report them as they appear to be. Some children indeed resolved oedema between admission and randomisation.

Line 113 and Table 2 – Why was the primary endpoint this composite enteropathy score, rather than a *change* in composite score from the time of enrollment in the study? (Or more ideally from the time of admission to hospital, as above?)

The composite score was selected from the Kosek paper (2014) because, although derived in stunted children rather than SAM, it had been generated in a large sample. To model the effects of the interventions, if any, we had a choice to calculate the change in each biomarker, or to model the final effect using the baseline value as a covariate. Our statistician chose the latter approach to modelling. Baseline data were not available at admission as we decided from the outset to recruit children only after stabilisation. This was because we were keen to reduce the adverse events and to avoid large numbers of deaths. Now that we are more confident in the safety of these interventions and we could do this in future work. We have clarified this in the text (lines 113-116).

Lines 115-116 and Table 2 – It would be fine to express that there is a trend towards improvement in the teduglutide group, but a p value of 0.07 doesn't quite meet traditional definitions of statistical significance.

This is absolutely true, but our intention here was to identify interventions which could be taken forward into further work. For this reason we pre-specified a less stringent hypothesis testing

threshold as at this stage it is critical to avoid a false negative trial. We do not wish to claim that any of these treatments are effective, just that they are worth further investigation and hence we would prefer to stick to the pre-specified significance level.

Line 119 – Same regarding the $p = 0.09$ value here. (Yes, I question the utility of p values too.)
As above

Line 127 – Same.
As above

Reviewer #3 (Remarks to the Author):

The study described in the manuscript by a research group from Zambia and Zimbabwe aimed to test the effect of four intestinal health targeting interventions in hospitalized children with severe malnutrition. These interventions consisted of bovine colostrum, N-acetyl glucosamine, budesonide and teduglutide and were tested using a phase II trial. Enteropathy is common in children with malnutrition and likely contributes to the high mortality rate in this population. There are no current interventions to facilitate restoration of intestinal health in malnourished children. Therefore, the trial is timely and of significant global health importance. The different interventions are well justified and the trial design is mostly appropriate. However, I do have a number of concerns which I have listed below.

A comment comment relates to the primary outcome. In the introduction, it is stated a broad range of outcomes were selected, but the primary outcome is based on three biochemical parameters in stool which is confusing and also not consistent with the BMJ protocol article (2019). In addition, the rationale for the stool composite score and the specific stool proteins selected should be better explained and their limitations discussed in detail. Limitations include falsely high levels due to upper respiratory infections leading to ingestion of neutrophils, dilution of stool samples due to diarrhea. In general, there is still uncertainty how well the stool biochemical readouts reflect severity of intestinal disease and this should be more explicitly discussed. This also relates to the validity of the composite score.

We cannot identify any discrepancy between the way this primary endpoint was calculated in this report and in the protocol paper. In the BMJ protocol article the primary outcome was described in detail (in the section “Outcomes and endpoints”, and the method of calculation was clearly set out in “Sample size and data analysis”).

We do agree that the interpretation of the primary endpoint is open to debate and have included a discussion of this important point in the Discussion (lines 213-221).

Adding a combination arm to this trial would have had added value as the interventions target different enteropathy pathways and there is a reasonable chance for synergistic effects. The lack of a combination arm should be discussed as a limitation.

This has been included in the Discussion (line 263).

There are important methodological details missing:

- Details on stool collection, sample transportation and storage. (page)
- Specifics on the ELISA's performed, (page)
- how the morphometrics was performed including number of sections or villi visualized. (page)
- rationale for specific dosing of the study drugs should be added. (page)
- IGFBP-3 seems to be an outcome but not described in the methods (page)

These have been included (pages listed above).

There are some inconsistencies between the published protocol manuscript in BMJ (2019) and the current manuscript. IGF-1 was listed as a secondary outcome, some of the morphometric data mentioned in the protocol paper is not

presented in the current manuscript. This data needs to be added or if not possible, the discrepancy needs to be clarified.

IGF-1 data are not presented because the values were almost all close to zero and we were concerned about quality. Insufficient plasma is available for re-testing, so we excluded the data. Other endpoints set out in the BMJ protocol paper but not presented in this manuscript are the lactulose-rhamnose ratio (performed on the endoscopic subset only and yielded only 13 data points), LPS and albumin (data failed QC test across countries).

The site specific differences in the primary outcome need to be part of the discussion as this could have implications for the generalizability. Can the authors speculate why children in these sites might respond differently and why there was no clear effect of any of the interventions on the primary outcome in the Lusaka site?

This has been included in the Discussion (lines 221-226).

Line 119 mentions a site specific effect presented in Suppl Table 2, but appears to be an error.

This has been corrected.

Summary of endpoints, drawn from protocol (version 3.0, page 26) showing which endpoints are included in the current manuscript

Primary Objective	Endpoint	Outcome Measures	Included in current manuscript?
Determine if interventions can improve malnutrition enteropathy	Recovery of mucosal integrity	Enteropathy biomarker score (faecal myeloperoxidase, alpha1-antitrypsin, neopterin)	Yes, this follows exactly what was laid out in the protocol and in the methods paper
Secondary Objectives	Secondary Endpoints	Outcome Measures	
Biomarker concentrations		Plasma fatty acid binding protein (FABP)	Yes, in Table 2
		Plasma lipopolysaccharide	No, as we were unable to resolve QC differences between study sites
		Plasma LBP	Yes, in Table 2
		Plasma soluble CD14, CD163	Yes, in Table 2
		Plasma C-reactive Protein	Yes, in Table 2
		Plasma albumin	No, as we were unable to resolve QC differences between study sites

		Hormones: GLP-2, IGF-1 and IGFBP3	Yes, in Table 2, except for IGF-1 values which were not included
Clinical course		Mortality by day 15 and day 28	Yes, in text (page 6)
		Adverse events, including clinical events and laboratory toxicity evaluated by full blood count and biochemistry	Yes, in text (page 6) and Supplementary Tables 3 and 4
		Serious adverse events	Yes, in text (page 6)
		Days with diarrhoea	Yes, in Table 2
		Days with fever	Yes, in Table 2
		Change in weight, weight-for-height and MUAC between baseline and day 15	Yes, in Table 2
		Days with oedema (children with oedematous SAM only)	Yes, in Table 2
Intestinal mucosal assessment (Lusaka only)		Villus height, crypt depth, villus width, epithelial surface area, inflammation score	Yes, in text
		Lactulose and rhamnase recovery	No, too few data points as LR testing was only done on the endoscopic subset and several collections failed so we ended up with only 15 paired (before/after) results

REVIEWERS' COMMENTS

Reviewer #1 (Remarks to the Author):

The authors have fully addressed my comments.

Reviewer #2 (Remarks to the Author):

The authors have satisfactorily addressed this reviewer's original comments and concerns.

Reviewer #3 (Remarks to the Author):

The authors have responded appropriately to the comments raised by different reviewers. I have no further comments.